# Grain Growth Mechanism of Lamellar-Structure High-Purity Nickel via Cold Rolling and Cryorolling during Annealing

**DOI:** 10.3390/ma14144025

**Published:** 2021-07-19

**Authors:** Zhide Li, Yuze Wu, Zhibao Xie, Charlie Kong, Hailiang Yu

**Affiliations:** 1Light Alloys Research Institute, Central South University, Changsha 410083, China; lizhide@csu.edu.cn; 2State Key Laboratory of High Performance Complex Manufacturing, Central South University, Changsha 410083, China; wyz2019@csu.edu.cn; 3College of Mechanical and Electrical Engineering, Central South University, Changsha 410083, China; xiezhibao@csu.edu.cn; 4Mark Wainwright Analytical Centre, University of New South Wales, Sydney, NSW 2052, Australia; c.kong@unsw.edu.au

**Keywords:** high-purity nickel, cryorolling, annealing, grain growth, lamellar-structure grains

## Abstract

High-purity (99.999%) nickel with lamellar-structure grains (LG) was obtained by room-temperature rolling and cryorolling in this research, and then annealed at different temperatures (75 °C, 160 °C, and 245 °C). The microstructure was characterized by transmission electron microscopy. The grain growth mechanism during annealing of the LG materials obtained via different processes was studied. Results showed that the LG high-purity nickel obtained by room-temperature rolling had a static discontinuous recrystallization during annealing, whereas that obtained by cryorolling underwent static and continuous recrystallization during annealing, which was caused by the seriously inhibited dislocation recovery in the rolling process under cryogenic conditions, leading to more accumulated deformation energy storage in sheets.

## 1. Introduction

Nanoscale lamellar-structure grains (LG) of nickel prepared by surface mechanical grinding treatment show ultrahigh strength and excellent thermal stability [1]. The molecular dynamics simulation results showed that LG nickel with the texture of {111} <110> and staggered low-angle grain boundaries was the most stable configuration [2]. Low-angle grain boundaries of LG are normally formed through dislocation multiplication and interaction during plastic straining [3]. Various severe plastic deformation techniques to produce LG material have been developed, such as surface mechanical grinding treatment [4,5], cryorolling [6,7], cross rolling [8], and other special rolling techniques [9,10]. The study on the grain growth mechanism of LG materials prepared by different processes with annealing can support further understanding of the thermal stability of LG materials. Moreover, a preparation procedure reference of LG materials with superb properties and exceptional thermal stability can be proposed in such studies.

During annealing, the grain growth of LG material is affected by many factors. The study of Liu et al. [10] showed that the high fraction of grain boundary microstructure characteristics with small angles contributes to the improvement of thermal stability for Ni alloys. Furthermore, the 50 °C delayed recrystallization starting temperature indicates that the recovery process of LG materials is affected by grain boundary orientation. Chen et al. [11] reported that the initial cubic fraction had a significant effect on the recrystallization process of a high-purity (99.999%) nickel plate in the cold rolling process with 96% reduction, indicating that the recovery of LG material was also influenced by the original grain size. Another study showed that, under the common influence of many factors, the recovery mechanism of LG materials varies [12]. Xie et al. [13] reported the preparation of a 99.945% pure Ni sheet with LG, with an average lamellar boundary spacing of 77 nm, via dynamic plastic deformation and cold rolling. During annealing, the local faceting of lamellar boundaries leads to structural coarsening and polygonization of lamellar structure. They reported a novel mechanism via coarsening of the initial structure of nanolaminates with low angle boundaries called local faceting [14]. Cryorolling is a special rolling technique widely used to prepare LG materials [6,7,15]. LG aluminum alloy with high strength and high ductility can be obtained by cryorolling with proper reduction and a subsequent annealing process [16]. However, the grain growth mechanism of cryogenic rolled Ni with LG has not been studied.

In this paper, high-purity (99.999%) nickel with LG was prepared by cryogenic rolling and room-temperature rolling. Then, the recovery characteristics and grain growth mechanism of lamellar grain structures prepared by both processes with annealing at different temperatures were studied.

## 2. Materials and Methods

The material used in this study was high-purity nickel (99.999%) to minimize the additional influence of impurities on grain boundary migration rate. The initial thickness of the material was 2 mm, which was thinned to 0.2 mm by room-temperature rolling and cryorolling. The relative reduction in each rolling pass was 10%, with a total reduction of 90% and equivalent strain of $\varepsilon_{\mathrm{VM}}=2.7$. The roll diameter of the rolling mill used in the rolling process was 80 mm, and the rolling speed was 1 m/min. Cryorolling was carried out in liquid nitrogen (−196 °C). The sample was kept in the liquid nitrogen for 30 min before the first pass of cryorolling to fully cool it. Then, the sample was kept in liquid nitrogen for 8 min before each pass, maintaining a cryogenic state throughout the process. Afterward, the rolled samples were annealed in a vacuum chamber at 75 °C, 160 °C, and 245 °C, with an annealing time of 1 h.

Tensile tests of the rolled and annealed samples were carried out on an AGS-X 10 kN tensile tester (Shimadzu, Kyoto, Japan) with a tensile rate of $1\times 10^{-3}\;s^{-1}$; the gauge length of the tensile samples was 18 mm, and the width of the tensile samples was 3 mm. The microstructure of the material was characterized by a Philips CM200 transmission electron microscope (TEM) (Philips Electron Optics, Eindhoven, The Netherlands) with a working voltage of 200 kV. The transmission sample was prepared using a focusing electron beam, and the observation surface was the cross-section of the rolling direction (RD) and the normal direction (ND). In order to obtain an accurate boundary spacing, it was quantitatively measured from the TEM images using the software Image J (ImageJ 1.53c, National Institutes of Health, Bethesda, Rockville, MD, USA). The final results were obtained by analyzing statistics collected from more than 100 grains.

## 3. Results

### 3.1. Microstructure Evaluation

The microstructure of pure nickel after room-temperature rolling with 90% reduction is shown in Figure 1a, in which the characteristics of LG can be clearly observed. It can be noted from the elongated grains with light and dark distribution that a large number of dislocations accumulated inside the grains. The microstructures of samples annealed at 75 °C, 160 °C, and 245 °C for 1 h are shown in Figure 1b–d. After annealing at 75 °C and 160 °C for 1 h, it can be observed from the TEM results in Figure 1 that the grain size did not increase significantly, indicating that the energy provided by annealing was not enough to drive the rapid grain boundary migration at the given temperature and holding time, even if dislocation recovery occurred. However, the grain was obviously coarsened after annealing at 245 °C for 1 h, as shown in Figure 1d. Almost pure white grains can be observed in the figure, indicating the presence of recrystallized grains. For recrystallized grains, only a few dislocations were observed near grain boundaries and within the grain, as indicated by the red arrow in the figure.

The microstructure of pure nickel after cryorolling with 90% reduction is shown in Figure 2a. It can be observed that the grain size was not significantly different from that after room-temperature rolling. This shows that the grain refinement effect of cryorolling is similar to that of room-temperature rolling under the same deformation. The microstructures of samples annealed at 75 °C, 160 °C, and 245 °C for 1 h after cryorolling are shown in Figure 2b–d. The cryorolled grain size did not increase significantly after annealing at 75 °C and 160 °C for 1 h. The grains of the cryorolled samples after annealing at 245 °C for 1 h were obviously coarsened; however, unlike the recrystallization of the room-temperature rolled sample, there were a large number of intersecting subgrain boundaries in the coarsened grains. As shown in Figure 2a, clear grain boundaries could be observed after cryorolling without annealing, while there were no intersecting subgrain boundaries, as shown in Figure 2d, indicating that these subgrain boundaries were formed during annealing.

For the better understanding of the changes in the microstructure of the samples rolled and annealed, quantitative measures were carried out on the grain boundary spacing of LG as shown in Figure 1 and Figure 2. Histograms were drawn according to the frequencies of the nine different levels, which were divided by the range of the measured statistics. Then, the mean grain boundary spacing was obtained by fitting curves of normal distribution to the frequency statistics, which are shown in Figure 3. It can be seen from Figure 3a,b that there was no obvious difference in grain boundary spacing of room-temperature rolling and cryorolling with 90% reduction. Severe plastic deformation forced the grain boundary facet to migrate, mainly by applying strong external stress as the mechanical driving force. Grains are constrained by neighboring grains during the deformation process; therefore, to achieve strain compatibility, numerous geometrically necessary dislocations (GND) need to be generated during the strain process between different grains, which separates the grains into fine grains, causing their refinement. With a 90% reduction, both room-temperature rolling and cryorolling required the materials to reduce the grain boundary spacing to adapt to macroscopic thinning, and the grain boundary spacing was similar after room-temperature rolling and cryorolling. There was a saturated grain size at a certain extent of grain refinement, depending on the deformation temperature, the initial grain size, the composition, and other factors affecting the dislocation accumulation and recovery. The size of the saturated grains could be indicated according to the Zener–Hollomon parameter, $Z=\overset{\cdot }{\varepsilon }\exp(Q/\mathrm{RT})$, where $\overset{\cdot }{\varepsilon }$ is the strain rate, Q is the diffusion activation energy in grain boundaries, R is the gas constant, and T is the deformation temperature [17]. The study of Pippan et al. [18] showed that the saturated grain size decreased significantly as the deformation temperature declined. This kind of dependence is stronger at the medium homologous temperature, but relatively weaker at the lower temperature. The study of Duan et al. [19] showed that, as the ARB cycles increased, the boundary spacing thickness of industrial purity (99.8%) nickel sheets decreased, reaching a saturation thickness of 75 nm after 6–8 cycles. In this paper, the grain boundary spacing of high-purity nickel reached 68 nm and 66 nm after room temperature rolling and cryorolling, respectively, which may have reached or been close to saturation thickness. In Figure 3c,e, the statistical results of grain boundary spacing after rolling at room temperature with annealing at 75 °C and 160 °C are shown, respectively. Figure 3d,f are the statistical results of grain boundary spacing of cryogenic rolling after 75 °C and 160 °C annealing. Both the statistical histogram and the fitting curve show that the grain boundary spacing after cryorolling and annealing was more evenly distributed with a smaller range fluctuation than that after room-temperature rolling and annealing, indicating that a more uniform microstructure was obtained with cryorolling compared to room-temperature rolling.

### 3.2. Mechanical Properties

The tensile curves of pure nickel after room-temperature rolling, cryorolling, and subsequent annealing are shown in Figure 4. The mechanical properties of pure nickel after room-temperature rolling, cryorolling, and subsequent annealing are shown in Table 1. It can be seen that the strength of the rolled samples remained high after annealing below 160 °C. On the other hand, the strength dropped significantly after 245 °C annealing with elongation increasing, which indicates that recrystallization occurred in the 245 °C annealing samples. The difference in tensile strength and elongation between the high-purity nickel samples obtained after room-temperature rolling and cryorolling, followed by annealing at 75 °C and 160 °C or no annealing, was little, as also shown by the grain size in the TEM results. However, the elongation of the samples annealed at 245 °C after cryorolling was better than that annealed at 245 °C at room temperature, which may be related to the different ways of grain growth in the process of annealing.

## 4. Discussion

### 4.1. Recovery Characteristics of LG Nickel Annealed at Low Temperature (75–160 °C)

The grain boundary spacing of LG nickel after room-temperature rolling and cryorolling was similar. After annealing at 75 °C and 160 °C, the grain boundary spacing of samples with different rolling processes began to differ. The grain boundary spacing changes of pure nickel annealed at 75 °C and 160 °C after cryorolling and room temperature rolling are shown in Figure 5. As can be seen from the figure (black symbols), the grain boundary spacing of the sample prepared by room temperature rolling with 75 °C annealing increased at a relatively fast rate as the annealing temperature increased. On the other hand, the grain boundary spacing of the sample prepared by cryorolling with lower temperature (75 °C) annealing (red symbols) increased at a rather slow rate. At the 160 °C temperature of annealing, the grain boundary distance of samples prepared by both rolling technique increased as the annealing temperature increased, while the increase speed of the room-temperature rolled sample was still obviously higher than that of the cryorolled sample. It is worth noting that the difference between the increase rate of grain boundary distance of room-temperature rolled samples and cryorolled samples decreased. This shows that the cryorolled samples had higher initial grain-coarsening temperature in the condition where grains had similar size. When annealed at lower temperatures (75 °C and 160 °C), the nickel sheets mainly underwent a static dislocation recovery process. For the room-temperature rolled samples, even in those with a faster grain boundary spacing increase rate, only 24 nm and 44 nm increments were observed after 75 °C and 160 °C annealing. The dynamic dislocation recovery process occurs during the process of room-temperature rolling on pure nickel, while it is strongly inhibited in the low temperature of cryorolling [20]. That is to say, room-temperature rolling is a process of continuous generation and annihilation of dislocation, whereas, in the cryorolling process, there is only continuous generation of dislocation, and the annihilation process is extremely suppressed, which leads to more dislocations accumulated during cryorolling compared to room-temperature rolling. When annealed at a lower temperature (75 °C), the dislocation recovery rate was slower, such that the cryorolled samples had more dislocations to recover from in the annealing process than after room-temperature rolling. Therefore, the grain boundary spacing increase rate of the cryorolled samples was lower than that after room-temperature rolling. When annealed with a higher temperature, the thermal effect increased the dislocation recovery rate, weakening the hindering effect of dislocation on the grain boundary spacing increment, leading to a narrowed gap between the grain boundary spacing increase rate of the cryorolled and room-temperature rolled samples.

### 4.2. Recovery Characteristics of LG Nickel Annealing at 245 °C

The microstructure shown in Figure 1d and Figure 2d, as well as the tensile results in Figure 4, reveal that, with annealing at 245 °C, both samples recrystallized. Figure 6 further shows the recrystallization behavior of the cryorolled and room-temperature rolled samples with subsequent annealing at 160 °C and 245 °C. With annealing at 160 °C, the room-temperature rolled samples showed local facet angularization in the lamellar boundary. As shown in the area marked by the yellow box in Figure 6a, the grain boundary facet of the B1 grain migrated toward the A1 grain region, exhibiting protruding sharp angles, as pointed out by yellow arrows in the figure. A similar phenomenon also existed between the B2 and A2 grains. Different phenomena were found in the LG boundary facets with cryorolling and annealing at 160 °C. As shown in the red box in Figure 6c, the grain boundary facet of D1 grain migrated to C1 grain while maintaining a state parallel to the rolling direction as a whole, as indicated by the red arrow in the figure. Moreover, the contrast of grains showed that the orientation difference between D1 and C1 grains was small, indicating that there may be subgrain boundary rotation with small grain boundary deviation during the grain boundary facet migration. TEM results of the sample rolled at room temperature with annealing at 245 °C are shown in Figure 6b. It can be seen that the dislocation density of grain B3 was low, while that of grain A3 was high. The results show that static discontinuous recrystallization occurred in the sample, and there was a process of grain nucleation to form grains with lower dislocation density before growing in the vicinity of the area with higher dislocation density. Figure 6d shows the TEM results of cryorolled samples annealed at 245 °C. As can be seen from the figure, the number of dislocations within grains decreased in the area indicated by red arrows, and the number of subgrain boundaries increased in the red boxes in the figure. These results indicate that continuous recrystallization may occur during annealing. Migration of the subgrain boundary was caused by dislocation slip and climb, leading to a transition to the large angle grain boundary. In terms of discontinuous recrystallization and continuous recrystallization, Jazaeri et al. conducted in-depth studies [21], which showed that small initial grain size and large strains would promote continuous recrystallization. The mechanism of continuous recrystallization is thought to involve the collapse and subsequent coarsening of the LG produced when rolling to large strains, which was consistent with our research. In the HRTEM results of Li et al. [22], it was intuitively observed that the incompletely recovered dislocation density of the cryorolled material with was is significantly higher than that of the material after room-temperature rolling and annealing. The LG obtained by cryorolling had a smaller initial grain size. The main reason was that cryorolling can accumulate dislocations more efficiently than room-temperature rolling. Even with the same amount of deformation, cryorolling was equivalent to room-temperature rolling with a larger amount of deformation. The results showed that the samples after cryorolling were more prone to continuous recrystallization.

An obvious difference in grain growth phenomenon during the annealing process in cryorolled and room-temperature rolled samples can be concluded from the discussion above. Due to the inhomogeneity of grain deformation, the grain with large deformation has a high dislocation density, while the grain with a small deformation has a low dislocation density. It is a common phenomenon that the original grain boundary arched nucleus and grows under the effect of deformation energy storage during the annealing process [23,24,25]. The schematic diagram of this process is shown in Figure 7a,b. However, there are obvious differences in the lamellar grains obtained by cryorolling during annealing. There is dislocation recovery in crystals, diffusion and rotation of grain boundaries, and the merging of subgrain boundaries into high-angle grain boundaries, resulting in the process of grain growth [26], as shown in Figure 7c,d. Li [27] proposed that there is less boundary migration caused by two subgrains merging into a larger grain, which are involved in the process of the subgrains rotating until the adjacent subgrains have similar orientations, whereby the two subgrains involved merge into a larger grain with less boundary migration, which is the driving force generated by the reduction in boundary energy. Because the dislocation recovery was seriously inhibited during deformation, the lamellar grain structure prepared by cryorolling had a high deformation energy storage, which may have been the reason for the subgrain boundary rotation and coalescence during annealing. Wang et al. [28] observed that nanocrystalline nickel could induce grain rotation and growth during deformation using in situ transmission electron microscopy. They pointed out that the rotation of grains can cause the reduction or elimination of differences in the neighboring grains’ orientation, thus leading to coalescence. Even without complete coalescence, the rotation of the entire grain may reduce the energy in the grain boundary system and effectively release stress concentration. When the grain size is small enough and the distortion of grain boundary is large enough, it is possible to release energy through grain boundary rotation during thermal activation.

## 5. Conclusions

In this paper, pure nickel was room-temperature rolled and cryorolled. The microstructures of the rolled samples and the annealed samples were observed by transmission electron microscopy (TEM). The grain growth mechanism during the annealing process of the high-purity nickel LG obtained by different rolling methods was studied, and the following conclusions were drawn:(1)Lamellar grain structures with a plane spacing of 68 nm and 66 nm were prepared by means of room-temperature rolling and cryorolling.(2)When annealed at 75 °C and 160 °C, the recovery rate of the samples after room-temperature rolling was faster than that after cryorolling.(3)After annealing at 245 °C, recrystallization occurred in both room-temperature rolled and cryorolled samples. Discontinuous static recrystallization occurred in the room-temperature rolled samples, whereas continuous static recrystallization occurred in the cryorolled samples; this was attributed to the greater deformation energy storage accumulated in the rolling process.

## Figures and Tables

**Figure 1 materials-14-04025-f001:**
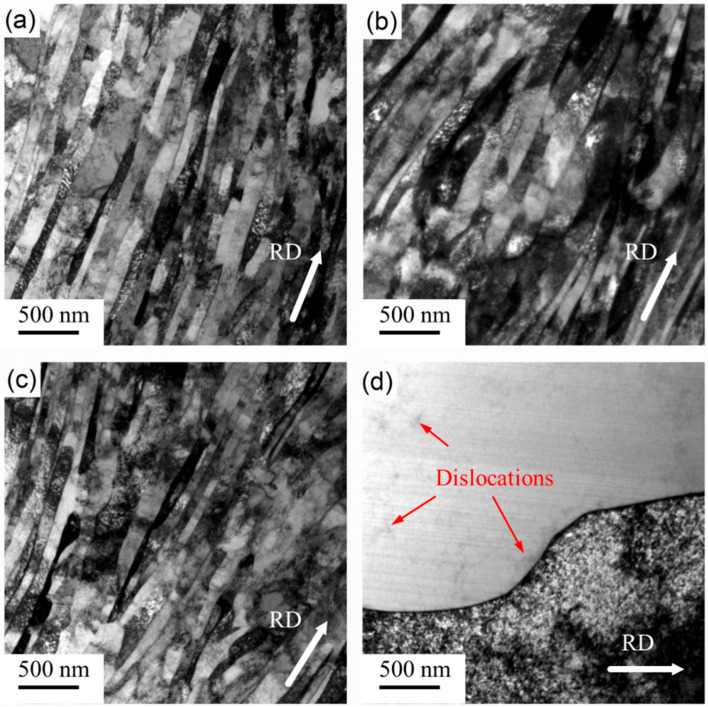
TEM images of Ni sheets subjected to (**a**) room-temperature rolling and subsequent annealing at (**b**) 75 °C, (**c**) 160 °C, and (**d**) 245 °C for 1 h.

**Figure 2 materials-14-04025-f002:**
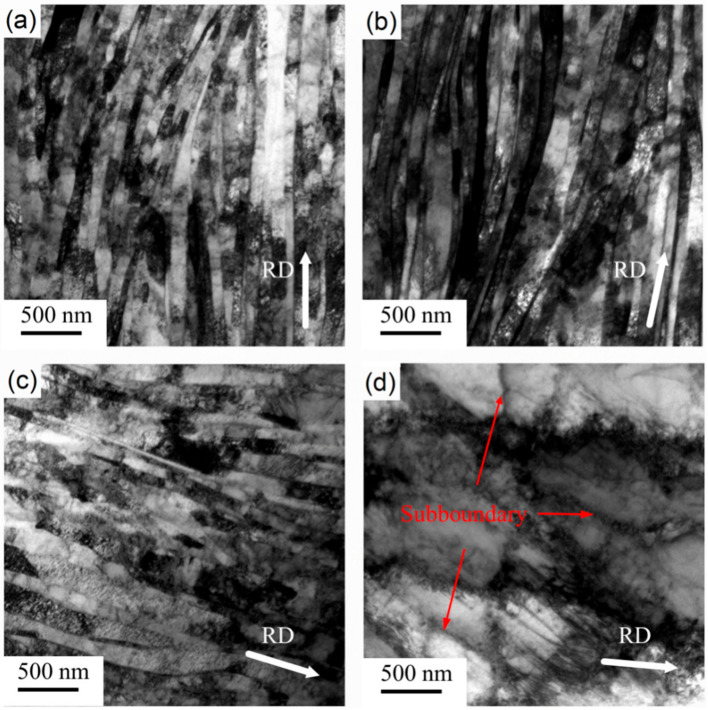
TEM images of Ni sheets subjected to (**a**) cryorolling and subsequent annealing at (**b**) 75 °C, (**c**) 160 °C, and (**d**) 245 °C for 1 h.

**Figure 3 materials-14-04025-f003:**
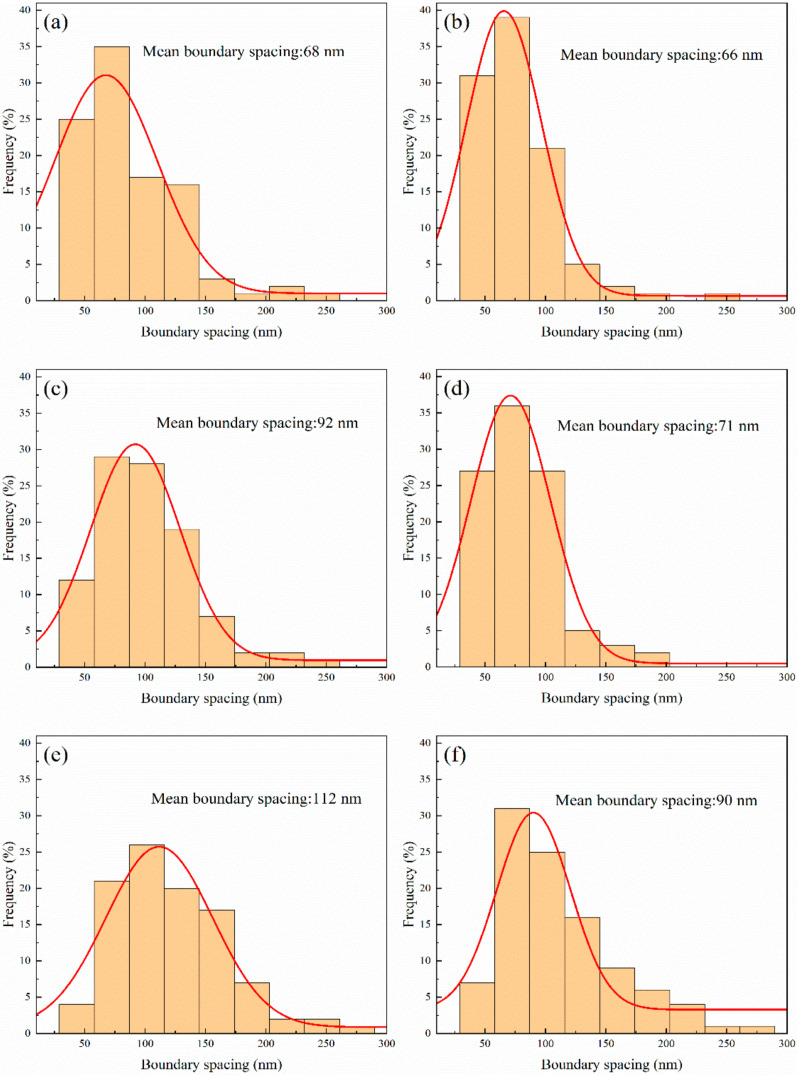
Histograms of boundary spacing of Ni sheets subjected to (**a**) room-temperature rolling and subsequent annealing at (**c**) 75 °C and (**e**) 160 °C for 1 h, and (**b**) cryorolling and subsequent annealing at (**d**) 75 °C and (**f**) 160 °C for 1 h.

**Figure 4 materials-14-04025-f004:**
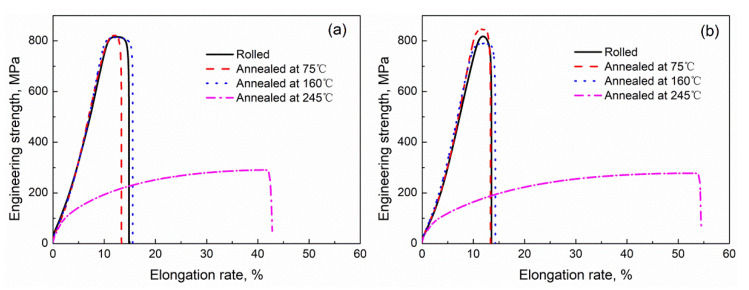
Tensile curves of sheets subjected to (**a**) room-temperature rolling and subsequent annealing, and (**b**) cryorolling and subsequent annealing.

**Figure 5 materials-14-04025-f005:**
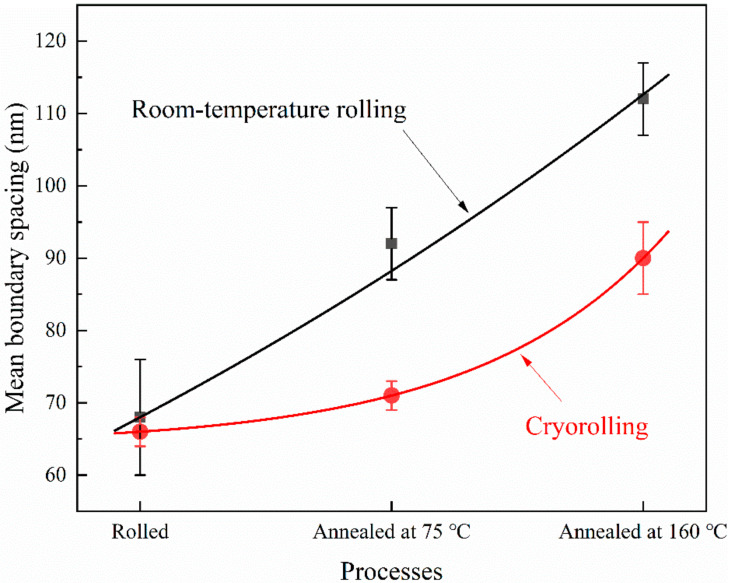
Mean boundary spacing of Ni sheets subjected to different processes.

**Figure 6 materials-14-04025-f006:**
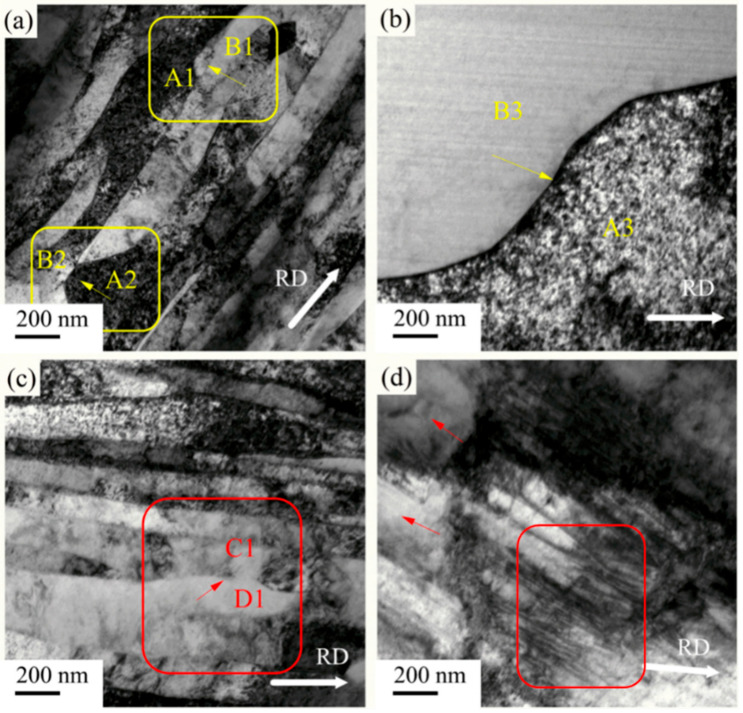
TEM images of the recrystallization behavior of Ni sheets subjected to room-temperature rolling with subsequent annealing at (**a**) 160 °C and (**b**) 240 °C, and cryorolling with subsequent annealing at (**c**) 160 °C and (**d**) 240 °C.

**Figure 7 materials-14-04025-f007:**
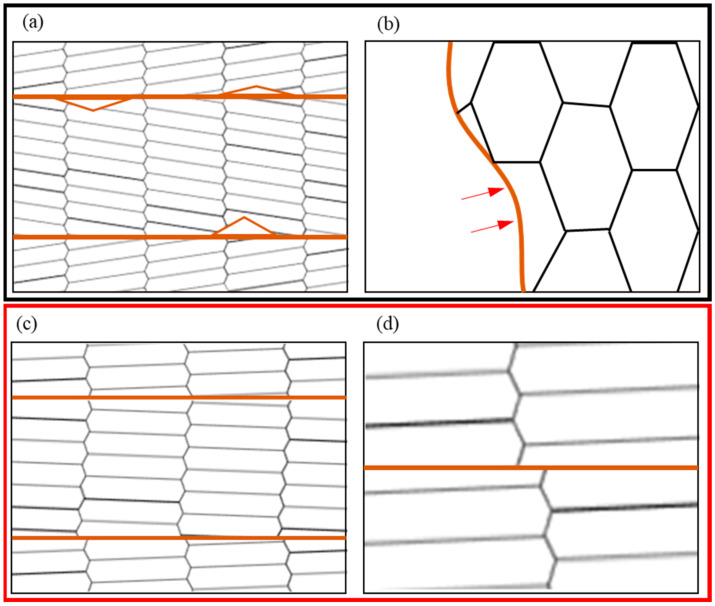
Schematic diagram of grain growth. (**a**,**b**) Discontinuous recrystallization processes: (**a**) serious lattice distortion and grain boundary arched nucleation; (**b**) grain growth. (**c**,**d**) Continuous recrystallization processes: (**c**) dislocations in grain recovery and subgrain boundary rotation; (**d**) slowly weakening lattice distortion with grain growth. The brown lines indicate subgrain boundaries and grain boundaries, and the black grid indicates the degree of lattice distortion.

**Table 1 materials-14-04025-t001:** Mechanical properties of pure nickel after room-temperature rolling, cryorolling, and subsequent annealing.

Processes	Annealed Temperature	Tensile Strength (MPa)	Elongation Rate (%)
Room-temperature rolled	As rolled	811 ± 3	14 ± 1
	Annealed at 75 °C	821 ± 6	14 ± 1
	Annealed at 160 °C	768 ± 11	14 ± 1
	Annealed at 245 °C	279 ± 10	48 ± 6
Cryorolled	As rolled	820 ± 2	14 ± 1
	Annealed at 75 °C	830 ± 10	14 ± 1
	Annealed at 160 °C	779 ± 8	14 ± 1
	Annealed at 245 °C	278 ± 9	54 ± 4

## Data Availability

The data presented in this study are available on request from the corresponding author. The data are not publicly available due to ongoing studies.

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
