# Peer review of "Grain Growth Mechanism of Lamellar-Structure High-Purity Nickel via Cold Rolling and Cryorolling during Annealing"

_materials, 2021, doi:10.3390/ma14144025_

Round 1

Reviewer 1 Report

In the manuscript titled "Grain growth mechanism of lamellar-structure high-purity nickel via cold rolling and cryorolling during annealing", the authors investigated the grain growth mechanism during annealing of the LG materials. 

I would like to thank the authors for this nicely written article. My only comment on the article is about the strains generated due to the orientation of the crystals. I would like to hear more discussion about differences in the lamellar grains obtained by cryorolling during annealing. If possible adding necessary discussion about the subject would increase the quality of the discussions section. 

Except for this point, I believe that this work will attract the interest of the readers researching the related topic.

Author Response

Thank you very much for your recognition of our work. And thank you for your kind suggestion. The first paragraph of section 4.2 of the discussion section has been revised: “In terms of discontinuous recrystallization and continuous recrystallization, Jazaeri et al. conducted in-depth studies [21], which showed that small initial grain size and large strains would promote continuous recrystallization. The mechanism of continuous re-crystallization is thought to involve the collapse and subsequent coarsening of the LG produced when rolling to large strains, which was consistent with our research. In the HRTEM results of Li et al.[22], it is intuitively observed that the incompletely recovered dislocation density of the cryorolled material with annealing is significantly higher than that of the material after room temperature rolling and annealing.The LG obtained by cryorolling had smaller initial grain size. The main reason was that cryorolling can accumulate dislocations more efficiently than room-temperature rolling. Even with the same amount of external deformation, cryorolling was equivalent to room-temperature rolling with larger amount of deformation. The results showed that the samples after cryorolling were more prone to continuous recrystallization.” An attempt was made to explain that the differences in the sample after cryorolling lead to continuous recrystallization during annealing.

[21] Jazaeri, H.; Humphreys, F.J. The transition from discontinuous to continuous recrystallization in some aluminium alloys II – annealing behaviour. Acta. Mater. 2004, 52, 3251-3262. doi:10.1016/j.actamat.2004.03.031.

[22] Li, J.; Gao, H.; Kong, C.; Tandon, P.; Pesin, A.; Yu, H. Mechanical properties and thermal stability of gradient structured Zr via cyclic skin-pass cryorolling. Mater. Lett. 2021, 302, 130406. doi: 10.1016/j.matlet.2021.130406.

Reviewer 2 Report

The work is interesting. The authors use a rather rare method of severe plastic deformation - cryorolling. However, there are some comments and suggestions fot improving the work (hereinafter, the numbers indicate the line in text):

74 - Was an extensometer used to determine elongation?

98 - Dimension line is hard to read, it is better to write it on a uniform background

169 - I would recommend adding a table with the obtained mechanical properties and dispersion. In addition, it would be nice to include in it the results obtained by other authors during cryorolling of nickel of varying purity, if it’s possible to find.

Author Response

  1. 74 - Was an extensometer used to determine elongation?

Answer: Thank you for your kind suggestion. No extensometer was used to determine elongation for the small size of the samples. In Fig. 4, “Engineering strain” have been changed to “Elongation rate”.

  1. 98 - Dimension line is hard to read, it is better to write it on a uniform background.

Answer: Thank you for your kind suggestion. The dimension line in Figure 1, Figure 2, and Figure 5 have been modified to make them easier to read.

  1. 169 - I would recommend adding a table with the obtained mechanical properties and dispersion. In addition, it would be nice to include in it the results obtained by other authors during cryorolling of nickel of varying purity, if it’s possible to find.

Answer: Thank you for your kind suggestion. “The mechanical properties of pure nickel after room temperature rolling, cryorolling and subsequent annealing are shown in Table 1.” and Table 1 had been added in Section 3.2. The mechanical properties of pure nickel cryorolling are seldom reported, so the results obtained by other authors during cryorolling of nickel of varying purity have not been included.